# Role of CYP3A5 in Modulating Androgen Receptor Signaling and Its Relevance to African American Men with Prostate Cancer

**DOI:** 10.3390/cancers12040989

**Published:** 2020-04-17

**Authors:** Priyatham Gorjala, Rick A. Kittles, Oscar B. Goodman Jr., Ranjana Mitra

**Affiliations:** 1Department of Biomedical Sciences, College of Medicine, Roseman University of Health Sciences, 10530 Discovery Drive, Las Vegas, NV 89135, USA; pgorjala@roseman.edu; 2Department of Population Sciences, Beckman Research Institute, City of Hope, Duarte, CA 91010, USA; rkittles@coh.org; 3Department of Internal Medicine, College of Medicine, Roseman University of Health Sciences, 10530 Discovery Drive, Las Vegas, NV 89135, USA; ogoodman@roseman.edu

**Keywords:** CYP3A5, androgen receptor, African American, CYP3A5 inhibitors/inducers

## Abstract

Androgen receptor signaling is crucial for prostate cancer growth and is positively regulated in part by intratumoral CYP3A5. As African American (AA) men often carry the wild type CYP3A5 and express high levels of CYP3A5 protein, we blocked the wild type CYP3A5 in AA origin prostate cancer cells and tested its effect on androgen receptor signaling. q-PCR based profiler assay identified several AR regulated genes known to regulate AR nuclear translocation, cell cycle progression, and cell growth. CYP3A5 processes several commonly prescribed drugs and many of these are CYP3A5 inducers or inhibitors. In this study, we test the effect of these commonly prescribed CYP3A5 inducers/inhibitors on AR signaling. The results show that the CYP3A5 inducers promoted AR nuclear translocation, downstream signaling, and cell growth, whereas CYP3A5 inhibitors abrogated them. The observed changes in AR activity is specific to alterations in CYP3A5 activity as the effects are reduced in the CYP3A5 knockout background. Both the inducers tested demonstrated increased cell growth of prostate cancer cells, whereas the inhibitors showed reduced cell growth. Further, characterization and utilization of the observation that CYP3A5 inducers and inhibitors alter AR signaling may provide guidance to physicians prescribing CYP3A5 modulating drugs to treat comorbidities in elderly patients undergoing ADT, particularly AA.

## 1. Introduction

Androgen depletion therapy (ADT) is the standard first line treatment in advanced prostate cancer [1]. Throughout the entire natural history of prostate cancer, AR remains active and is still expressed in patients undergoing ADT [2,3,4]. Mutated AR often can bypass the need for androgen activation, and can act as transcriptional activator in the absence of androgens, promoting tumor growth [5]. Several new therapeutic approaches are available to AR signaling, one of them being blocking non-gonadal androgen synthesis [6]. Nonetheless, eventually the AR bypasses these strategies, leading to CRPC. Identification of novel mechanisms to block AR nuclear translocation represents an unmet need [7,8,9].

Our previous work shows that CYP3A5 has a critical role in AR signaling as it promotes AR nuclear translocation and downstream signaling promoting growth [10]. CYP3A5 is a cytochrome P450 enzyme primarily expressed in liver and small intestine. In liver, its main function is to process xenobiotics. CYP3A5 along with CYP3A4 metabolizes 50% of the commonly prescribed drugs [11]; these drugs are inducers, inhibitors, and substrates of CYP3A enzymes. Apart from liver and small intestine CYP3A5 is also expressed in prostate where its normal function is to convert testosterone to its lesser active derivative, 6β-hydroxytestosterone [12]. Prostate cancer patients are typically elderly as the average age at diagnosis is 66 [13] and often suffer from comorbidities. Medications prescribed for these comorbidities can be inducers, inhibitors, or substrates of CYP3A5 and hence can modify intratumoral CYP3A5 activity and alter AR signaling and response to ADT. Prior reports strongly support our hypothesis that therapeutic management of cancers is compromised by drug-induced expression of members of the CYP3A subfamily [14]. CYP3A5 is the main isoform expressed in prostate, whereas CYP3A4 is the most common isoform expressed in liver. Although CYP3A4 and CYP3A5 share 80% similarity, they are differentially regulated [14,15,16]. CYP3A5 expressed in prostate is also differently regulated compared to the one expressed in liver as it has a 5’ UTR with androgen response elements (ARE). 

Healthy prostate epithelia are shown to express high basal levels of CYP3A5, but CYP3A5 expression in prostate cancer tissues is less well-characterized [17,18,19]. Different expression patterns in tumor cells may be due to polymorphic expression of CYP3A5. CYP3A5 has several variations most common being the CYP3A5*3, which carries a A>G mutation at position 6986 in the intron 3 (CYP3A5*3, rs776746 A>G). The presence of CYP3A5*3 results in aberrant splicing producing truncated non-functional protein [20]. Ninety percent of Non Hispanic White Americans (NHWA) carry the *3 mutation (*3/*3), whereas African Americans mostly carry (72%) the wild type CYP3A5 (*1/*1 or *1/*3) expressing the full-length protein. 

Previously we demonstrated that CYP3A5 facilitates nuclear translocation of androgen receptor in prostate cancer cells [10]. We have also demonstrated that CYP3A5 specific inhibitor, azamulin, and siRNA-based knock down of CYP3A5 expression reduced AR nuclear translocation. In the current study, we investigated the effect of commonly co-prescribed CYP3A5 inhibitors/inducers with androgen deprivation therapy on AR translocation and their modulation downstream signaling. This study may provide clinical guidance regarding optimal selection of CYP3A5 modulators to co-prescribe with ADT. As African Americans mostly express the full length CYP3A5 that promotes androgen receptor signaling and promotes prostate cancer growth, this study is very relevant to the AA patients that often have clinically aggressive diseases and may help to address interracial health disparities

## 2. Results

### 2.1. Differential Expression of CYP3A5 between African American and Non-Hispanic White Americans Origin AR Positive Prostate Cancer Cell Lines

We have previously shown that CYP3A5 is expressed in androgen receptor positive prostate cancer cell lines (LNCaP, C4-2 and 22RV1) and promotes activation of AR and prostate cancer growth [10]. CYP3A5 expression is polymorphic and is race linked, so we genotyped the available AR positive cell lines from both African American (AA origin, MDAPCa2b, RC77 T/E (Tumor), RC77 N/E (normal) and Non-Hispanic White Americans (NHWA-LNCaP, C4-2, 22RV1, E006aahT) origin to determine their CYP3A5 polymorphism. (Table 1, Appendix A). Genotyping revealed that all the NHWA lines carry the *3/*3 CYP3A5 variant in homozygous form. The three cell lines from AA origin, carry *1/*3 heterozygous wild type/mutant CYP3A5. E006aahT has been found to be not of African American origin [21] and carries *3/*3 homozygous mutation. We used LNCaP (*3/*3) and MDAPCa2b (*1/*3) cells for our current study as they are of NHWA and AA origin, respectively, and are AR positive commercially available (ATCC) and show similar response to androgens. Both LNCaP and MDAPCa2b express similar level of CYP3A5 mRNA but MDAPCa2b expresses higher levels of CYP3A5 protein (Appendix A).

### 2.2. CYP3A5siRNA Downregulates AR Nuclear Translocation in MDAPCa2b Cells Expressing Wild Type CYP3A5 (*1/*3)

To test if wild type full length CYP3A5 regulates AR nuclear activation in a similar fashion, we used MDAPCa2b cells, which express wild type CYP3A5 (*1/*3). MDAPCa2b cells were treated with NT and CYP3A5 siRNA pool to specifically block CYP3A5 and then induced with DHT in charcoal stripped phenol red free media to monitor AR nuclear translocation and activation. The cytoskeletal fraction, which contains CYP3A5, shows reduced CYP3A5 protein after siRNA treatment as compared to non-target (NT) control (Figure 1A). Of note, CYP3A5 siRNA treatment did not affect total AR protein expression (Figure 1A). Cells were then stained with AR and Cy5 labelled secondary antibody. CYP3A5 siRNA treatment resulted in decreased nuclear translocation of AR (Figure 1B). This observation was further confirmed with cell fractionation experiments performed after NT and CYP3A5 siRNA treatment and DHT induction. Western blotting analysis was performed to monitor AR nuclear translocation in the cytoplasmic and nuclear fractions. The result confirms our previous observation with the LNCaP cell line—CYP3A5 siRNA treated MDAPCA2b cells show decreased nuclear translocation of AR (Figure 1C) after DHT induction as compared to non-target siRNA pool treated cells. In the NT siRNA treated group, we observed significant cytoplasmic surge, which was absent in the CYP3A5 siRNA treated cells. 

### 2.3. CYP3A5 siRNA Downregulates Expression of AR Regulated Genes in MDAPCA2b Cells

To further evaluate the downstream signaling effect of CYP3A5 knockdown, cDNA prepared with RNA extracted from MDAPCA2b cells treated with NT and CYP3A5 siRNA pool was used for gene expression analysis. RT^2^ PCR pathway array deciphering changes in signaling targets downstream of Androgen receptor shows downregulation of several genes listed in Table 2 with fold changes greater than 2.0 and p-values less than 0.005 depicted (Figure 2A). The effectiveness of the CYP3A5 siRNA pool was confirmed using western analysis and confocal microscopy (Figure 2B and Appendix A). Western analysis was performed to evaluate whether gene expression changes translated into changes in protein expression. FKBP5, c-Myc, ELK-1, prostein protein expression also decreased in response to CYP3A5 siRNA treatment (Figure 2B), consistent with mRNA downregulation (fold change 0.68, 0.55, 0.49 and 0.45, respectively, P value ≤ 0.05). We did not observe changes in the levels of MME, SPDEF, and KLK2 protein levels with CYP3A5 siRNA treatment (Figure 2C).

### 2.4. Commonly Co-Prescribed CYP3A5 Inducers/Inhibitors can Alter AR Nuclear Translocation

The average age at detection is 66 for prostate cancer patients; hence, they often have comorbidities, and are prescribed other medications to treat these co-morbidities, while undergoing androgen deprivation treatment (ADT). CYP3A5 is known to process 33% of the commonly prescribed drugs and these co-prescribed drugs can be an inducer/inhibitor of CYP3A5. Since AR is central to prostate cancer progression and is a main therapeutic target in treating prostate cancer, any alteration in AR signaling can alter efficacy of these regimens. Based on our observation that CYP3A5 alters AR activity, we wanted to test the effect of CYP3A5 inducers/inhibitors drugs on AR signaling, as CYP3A5 can be modulated by these co-prescribed drugs. To evaluate the effect of known CYP3A inducers and inhibitors on AR nuclear translocation and downstream signaling, we used two CYP3A inhibitors, amiodarone (5 µM) and ritonavir (35 µM); and two inducers, phenytoin (50 µM) and rifampicin (30 µg/mL) [22]. Amiodarone is often prescribed as an anti-arrhythmic drug, whereas ritonavir is a component of highly active anti-retroviral therapy used in treating HIV patients. Phenytoin is a commonly prescribed antiepileptic drug and rifampicin is an antibiotic and known CYP3A5 inducer. We tested their ability to affect AR activation process due to their ability to modulate CYP3A5 expression, which is separate from their primary target. Azamulin, a specific CYP3A inhibitor has been used as a control.

We tested the effect of these CYP3A5 inhibitors and inducers on total AR expression. Cell lysates prepared from LNCaP and MDAPca2b cells incubated with the selected drugs were analyzed by western blotting to verify if protein expression of AR is affected. None of the drugs tested affects total AR protein expression in LNCaP and MDAPCa2b cells, except ritonavir (Figure 3A). 

We observed reduced AR nuclear translocation in the cells treated with CYP3A inhibitors (amiodarone and ritonavir) and increased AR translocation in cells treated with CYP3A inducers (phenytoin and rifampicin) as compared to control cells that received no drugs (vehicle treated) (Figure 3B) in both LNCaP and MDAPCa2b cell lines expressing different levels of CYP3A5 full length protein. Additionally, the cells treated with CYP3A inducers showed increase nuclear AR even without DHT induction compared to control. Similarly, the CYP3A5 inhibitor treated cells show lower nuclear AR also without DHT induction in both LNCaP (*3/*3) and MDAPCA2b (*1/*3) cells. We confirmed the CYP3A5 modulating effect of these drugs on AR activation by performing cell fractionation studies with and without DHT induction.

### 2.5. Changes in AR Activation by CYP3A Inducers are due to Their Effect on CYP3A5 Activity

To test our hypothesis that AR nuclear localization is dependent on the changes in CYP3A5 expression caused by the CYP3A5 inducers/inhibitors, we performed AR nuclear localization assays after CYP3A5 and NT siRNA treatment. The MDAPCa2b cells treated with CYP3A5 siRNA and CYP3A5 inducer (phenytoin and rifampicin) do not show increased nuclear AR in contrast to NT control (Figure 4). This result supports that the observed changes in AR nuclear fraction is dependent on the modulation of CYP3A5 by the above-mentioned CYP3A5 inducers (rifampicin and phenytoin) and is independent of their effect on the main primary target.

### 2.6. CYP3A5 Inhibitors and Inducers Alter PSA Levels

Prostate specific antigen (PSA) expression is regulated by androgen receptor and is an established marker to monitor AR downstream signaling. To evaluate downstream effects of AR nuclear translocation due to pharmacologic modulation of CYP3A5 expression, we analyzed the level of PSA protein expression in the phenytoin, rifampicin, and amiodarone treated cells. In both LNCaP and MDAPCa2b cell lines, CYP3A-inducing drugs phenytoin and rifampicin increased expression of PSA. The fold change in the MDAPCa2b cell line, which carries a wild type CYP3A5 (Figure 5A), was more compared to the LNCaP line carrying mutant CYP3A5 (*3/*3). As expected, amiodarone reduced PSA protein expression in both the cell lines; the effect is more prominent after 48 hours of DHT treatment. Since the primary target of these drugs is not CYP3A5, we tested the effect of one of these drugs, namely phenytoin, in a CYP3A5 knockout background. In the absence of CYP3A5, we do not see increased PSA production, as observed in the case of NT control (Figure 5B). This difference is more prominent after DHT induction, indicating that it is dependent of phenytoin’s effect on CYP3A5 (induction), which is known to promote AR activation and downstream signaling.

### 2.7. CYP3A5 Modulating Drugs Affect AR Downstream Signaling 

We used a luciferase-based reporter assay to determine the effect of commonly prescribed CYP3A5 inhibitors/inducers on their ability to modify AR downstream signaling. Both LNCaP and MDAPCa2b cell lines were transduced with a viral construct carrying androgen response elements (AREs) fused with luciferase; positive clones were selected after antibiotic selection. Negative controls were setup with constructs carrying only the TATA promoter without ARE. A pool of positive clones was used to monitor changes in luciferase activity after treatment with the CYP3A5 inducers/inhibitors. The reporter assay using MDAPCA2b cells show increased luciferase activity with CYP3A5 inducers (phenytoin, rifampicin, and hyperforin) and decreased luciferase activity with inhibitors (ritonavir, amiodarone, and chloramphenicol) (less AR activation) (Figure 6A). LNCaP cells showed increased luciferase activity after treatment with CYP3A5 inducers phenytoin and rifampicin with DHT treatment; phenytoin shows an increase in AR activity even without DHT induction similar to earlier observation (Figure 6B). The inhibitors amiodarone and ritonavir show reduced luciferase units (AR activity) both with and without DHT induction (Figure 6B).

### 2.8. CYP3A can Regulate PCa Cell Growth by Modifying AR Activation

Androgen signaling pathway is involved in cell growth; based on our observation that CYP3A inhibitors and inducers alter AR nuclear translocation, we hypothesized that they should also alter cancer cell growth. To test our hypothesis, we monitored the effect of these inhibitors and inducers on prostate cancer cell growth. Both LNCaP and MDAPCa2b cell lines were incubated with different dose range of inducers (phenytoin (0–60 µM), rifampicin (0–35 µM)] and CYP3A inhibitors [amiodarone (0–6 µM), ritonavir (0–40 µM)). Our results indicate that CYP3A inhibitors amiodarone and ritonavir decreased cell growth whereas CYP3A inducers phenytoin and rifampicin reduce cell growth of both cell lines increasing concentrations (Figure 7). The effect of CYP3A inducers and inhibitors are more pronounced in MDAPCa2b cells compared to LNCaP, which may be due to the presence of wild type CYP3A5 (*1/*3), which has 3-4 times more functional CYP3A5 as compared to LNCaP (*3/*3).

## 3. Discussion

Our previous work shows that CYP3A5 inhibition can lead to growth inhibition in LNCaP cells due to blocking of AR activation and downstream signaling. In keeping with previously published results for LNCaP, the MDAPCa2b, which carries one copy of wild type CYP3A5 (*1), also promotes AR nuclear localization. CYP3A5 is polymorphic with the wild type variant encoding full length translated protein being expressed in 73% of AAs, whereas only 5% of this variant is expressed in NHWA [20,23]. Since *3 is the most common difference between AA and NHWA, we analyzed the available prostate cancer cell lines and used one (*3/*3, LNCaP) and the other (*1/*3, MDAPCa2b) cell line for this study. There are 12 known SNPs in the CYP3A5 gene that mostly result in inactive protein. Distribution of these SNPs between races varies depending on the SNPs. The most commonly expressed mutation (*3) is a point mutation at 6986A > G that results in alternative splicing of an insertion from intron 3 resulting in a nonsense-mutated nonfunctional truncated protein and is present in 95% of NHWA, whereas 75% of AA carry wild type and 10-13% of AAs carry *6 and *7 mutations (truncated protein) [24,25]. Even though A>G mutation leads to truncated protein in *3 mutation, 5% of the matured RNA can bypass the alternative splicing and express low levels of full length CYP3A5 protein as observed in LNCaP cells (*3/*3). Prevalent expression of wild type CYP3A5 (*1/*1) form can promote AR activation in the AA prostate cancer patients as compared to NHWA. Since CYP3A5 is the major extrahepatic CYP3A isoform expressed in prostate and regulates AR activation, the presence of these SNPs in CYP3A5 may alter prostate cancer occurrence growth and treatment resistance in a race-dependent manner. 

Since MDAPCa2b carries a wt CYP3A5, we used this cell line for the PCR based pathway array to study the effect of CYP3A5 inhibition on AR downstream signaling. The 11 genes that show maximum fold change (≥ 2.5) with CYP3A5 siRNA treatment are known to play an important role in prostate cancer growth and severity. SLC45A3, also known as prostein, is downregulated (−4.56 fold) with CYP3A5 siRNA treatment and belongs to solute carrier family 45. Protein expression is seen in both normal and malignant prostate tissue; its messenger RNA and protein are upregulated in response to androgen treatment in prostate cancer cells. [26,27]. FKBP5 (downregulated, −4.43 fold, also called FKBP51) is a co-chaperone that belongs to a family of immunophilins, FK506 binding proteins (FKBPs). FKBP5 works with several different signaling pathways, including steroid receptor signaling, NF-κB, and AKT pathways, all of which contribute to tumorigenesis and drug resistance [28,29] and FKBP5 is a target for AR signaling [30]. A recent study uncovered a mechanism in which FKBP5 is found to form a complex with HSP90 and promote AR signaling in prostate cancer [31]. Members of this family are targets for drugs such as rapamycin and cyclosporine. FKBP5 is known to modulate steroid receptor (androgen, progesterone, glucocorticoid) function by forming complex with HSP90 and HSP70. c-MYC, also significantly downregulated with CYP3A5 siRNA treatment, is one of the key genes amplified in prostate cancer progression. c-MYC induces AR gene transcription and is frequently upregulated in CRPC. A positive correlation between c-MYC and AR mRNA has been reported [32,33,34,35,36]. ELL2 (elongation factor, RNA polymerase II) is encoded by an androgen-response gene in the prostate [30,37]; it suppresses transient pausing of RNA polymerase II activity along the DNA strand and facilitates the transcription process [38]. ELL2 has been identified as an androgen response gene in immortalized normal human prostate epithelial cells as well as prostate cancer cell lines LNCaP and C4-2 [30,39]. ELL2 downregulation is seen in prostate cancer specimens and other observations indicate that its decrease improves cell proliferation, migration, and invasion [40]. However, another study by Zang et al. indicates that ELL2 has an important role in DNA damage response and repair. This enables ELL2 loss to function like a double edge sword which on one hand can induce prostate carcinogenesis, and on the other can sensitize cells to radiation therapy [41]. Human kallikrein-related peptidase 2 (KLK2, previously known as hK2) is a secreted serine protease from the same gene family as PSA. It shares 80% sequence homology with PSA and is responsible for cleavage of pre-PSA to active mature PSA [42]. Our studies only indicate fold change in mRNA levels but not protein levels after CYP3A5 siRNA treatment. Studies give contradictory evidence towards KLK2’s use as a marker for detection of prostate cancer in combination with PSA [43,44,45]. KLK2 has been found to modulate AR to increase cell growth after development of CRPC [46]. In conclusion, this data supports our earlier observation that CYP3A5 plays a major role in AR regulation, thus modulating AR downstream signaling and prostate cancer growth. This also points out how the presence of a wild type CYP3A5 (preferentially present in AAs) can significantly alter AR signaling compared to cells carrying only inactive CYP3A5 polymorphic forms (expressed in NHWAs).

CYP3A5 is an enzyme whose activity can be physiologically altered by many prescribed drugs that are activators or inhibitors of CYP3A5. Men with prostate cancer undergoing ADT are often elderly and have comorbidities requiring concomitant prescription medications, many of which are CYP3A5 inducers or inhibitors. The modulation of CYP3A5 by concomitant prescribed drugs may enhance or interfere with ADT, of great relevance to the AAs expressing wild type CYP3A5. Our data show that commonly prescribed CYP3A5 inducers promote AR nuclear migration, whereas CYP3A5 inhibitors block AR nuclear migration. The results of the study indicate that the CYP3A5 inhibitors show less nuclear AR and less PSA expression similar to CYP3A5 siRNA. Conversely, the inducers promoted nuclear AR translocation with and without DHT induction. Both the cell lines show similar effect since both they have different AR and CYP3A5 expression; we were not able to derive a quantitative difference between both the cell lines. Luciferase reporter assays showed a concordant response with respect to AR downstream signaling. Although CYP3A5 is not the intended target of any of these drugs, it was shown that a predictable effect on AR signaling is due to the changes in CYP3A5 and not due to the primary target of these drugs (Figure 4). The specificity of the effect of CYP3A5 inducer phenytoin was also tested in a CYP3A5 knockout background wherein it was not able to induce PSA production as observed with NT siRNA treatment (Figure 5B). These drugs inhibit both CYP3A4 and CYP3A5 isoform. Since CYP3A5 is the major extrahepatic form expressed in prostate, the observed effect on AR signaling is due to the alteration in CYP3A5 and not CYP3A4. 

Both CYP3A inducers increase the proliferation of the cells (LNCaP and MDAPCa2b) and inhibitors reduce cell growth. Interestingly the effect of inducers and inhibitors on growth are more pronounced in MDAPCa2b, which carries the wild type CYP3A5, as compared to LNCaP(*3/*3), with the exception of ritonavir. The observed difference can be because the other three tested inhibitors only effect the AR nuclear localization, whereas ritonavir also affects total AR levels. Although CYP3A5 inducers and inhibitors are well known to contribute to drug-drug interactions, a potential mechanism that may impact androgen receptor signaling has not been suggested or demonstrated previously. These data strongly suggest that concomitant CYP3A5 inhibitor/inducers prescribed to patients undergoing ADT may alter the efficacy of ADT. Although our data are highly suggestive and provocative, we recognize that they are preliminary proof of concept and may not be generalizable in the clinical setting. Prospective clinical studies are further needed to determine the clinical impact of concomitant prescription medications in men with prostate cancer undergoing ADT.

## 4. Materials and Methods 

### 4.1. Cell Lines, Drugs, and Antibodies

LNCaP, MDAPCa2b, 22RV1, and E066AAhT cells were purchased from ATCC and maintained in RPMI (Invitrogen, Carlsbad, CA), F-12K medium (ATCC® 30-2004), RPMI, and DMEM (Invitrogen, Carlsbad, CA, USA) media, respectively. Supplements were added as recommended by ATCC. C4-2 was a gift from Dr. David Nanus and maintained in RPMI media. RC77 T/E (tumor) and RC77 N/E (normal) cell lines were obtained from Dr. Johng S Rhim and are maintained in Keratinocyte SFM media supplemented with epidermal growth factor and bovine pituitary extract [47].

Antibodies against Androgen receptor (ab74272), were obtained from Abcam, (Cambridge, MA). Anti-CYP3A5 (MA3033) was from Thermo Fisher (Carlsbad, CA, USA) and anti-GAPDH (10R-G109A) was from Fitzgerald Industries (Acton, MA, USA). Anti-α tubulin (2125S) and anti Lamin A/C (4C11) were obtained from Cell Signaling Technologies (Danvers, MA, USA). The secondary antibodies (IR dye 680 and IR dye 800) were from LI-COR (Lincoln, NE, USA).

CYP3A inducers phenytoin (PHR1139) and rifampicin (R3501); and inhibitors ritonavir (SML0491), amiodarone hydrochloride (A8423), and azamulin (SML0485) were purchased from Sigma-Aldrich (St. Louis, MO, USA). 

The siRNA transfection reagents, oligofectamine, and OPTIMEM were from Invitrogen; the siRNAs were from Dharmacon (Thermo Scientific, Carlsbad, CA, USA). MTS cell titer reagent was from Promega (Madison, WI, USA).

### 4.2. Western Blotting 

Cells were washed in phosphate buffer (PBS) and lysed in RIPA buffer (50 mM TIRS, 150 mM NaCl, 1% NP40, 0.5% sodium deoxycholate, 0.1% SDS, 2 mM EDTA, 2 mM EGTA) supplemented with protease and phosphatase inhibitors for total protein isolation. GAPDH was used as an internal control for total protein. Infrared fluorescent-labeled secondary antibodies were used for detection using Odyssey CLx. All the densitometry analysis and fold change calculations are shown in Appendix A.

### 4.3. Cell Fractionation 

Nuclear and cytoplasmic cell fractionation was prepared using a NE-PER Nuclear and cytoplasmic extraction kit from Thermo Scientific (Cat No. 78833) and the manufacturer’s instructions were followed. The cells were treated with drugs in charcoal-stripped phenol red free media 48 hours after plating. Drugs and DHT were added at specified concentration and duration as indicated. Cells were washed in PBS once before the cells were suspended in CER buffer. Protease, phosphatase inhibitors, and EDTA was added prior to cell lysis. The pellet remaining after cytoplasmic isolation was washed twice with PBS. The pellet was suspended in NER buffer for nuclear fraction extraction according to guidelines and the samples were stored at −80°C until further processing. Cytoskeletal fraction was prepared using the Qproteome cell compartment kit from Qiagen LLC (Germantown, MD, USA). The cytoskeletal fraction was dissolved in CER 4 buffer as instructed and used for western analysis of CYPA5. Tubulin was used as internal controls for cytoplasmic and cytoskeletal fractions and lamin was used as a control for nuclear fractions.

### 4.4. siRNA Inhibition

The cells were plated in complete media without antibiotics on poly D-lysine-coated plates (80,000 cells per 6 well). After 48 h of growth, the cells were transfected using RNAimax according to the manufacturer’s instructions. The smart pool non-target (NT) siRNA (Dharmacon catalog# D-001810-10) was used as a transfection control with the experimental target gene siRNAs. A pool of four siRNA (Dharmacon catalog# L-009684-01) against the CYP3A5 were used to block the expression. The final concentration of the siRNA (NT and targets) used was 30 nM. 

### 4.5. Confocal Microscopy 

Cells were seeded into a 35-mm glass bottom dish (Cellvis catalog# D35C4-20-1.5-N). The cells were fixed in 4% paraformaldehyde for 20 minutes and permeabilized using a permeabilizing buffer (0.2% Tween 20 in PBS) for 5 minutes. The cells were blocked using 10% goat serum diluted in a permeabilizing buffer with 1% BSA for 15 minutes. Primary antibodies were diluted at 1:100 in staining buffer (1% BSA in PBS) and incubated for 60 minutes at room temperature. The cells were washed three times (10 minutes each) in PBS. Secondary antibodies, Cy^5^-conjugated Donkey Anti-rabbit (711-175-152), and Alexa Fluor 488—conjugated Donkey Anti-Mouse (715-545-150) from Jackson Immuno Research, West Grove, PA, were diluted at 1:50 in a staining buffer and incubated for 30 minutes at room temperature. The cells were washed three times (10 minutes) in PBS and stained with 1 µg/mL DAPI (4’, 6-diamidino-2-phenylindole) in PBS for 5 minutes at room temperature. The cells were stored in PBS at 4°C until imaging was completed.

The cells were imaged using confocal laser scanning microscopy on a Nikon A1R using a Galvano scanner and a 60× Apo-TIRF oil immersion objective. To excite DAPI, FITC, TRITC, and CY^5^ 405 nm, 488 nm, 561 nm, and 638 nm solid-state lasers were used, respectively. FITC and TRITC emissions were collected using GaAsP detectors on the A1R+ microscope. NIS-Elements software from Nikon was used for recording the data.

### 4.6. qPCR and RT2 Profiler Assays

RNA was isolated using the RNeasy Mini kit (Cat no.74104) from Qiagen (Germantown, MD, USA) using the manufacturer’s instructions. cDNA synthesis was performed using RT^2^ first strand kit (Cat No. 330404) Qiagen, according to the manufacturer’s instructions. The cDNA was diluted and used as template to analyze for gene expression pattern using individual gene assay or RT² Profiler PCR Arrays (Cat No. 330231) specifically designed to probe panel of Human Androgen Receptor Signaling Targets (PAHS-142Z). The real-time PCR reaction data was collected using ABI 7500 fast real-time PCR system. For CYP3A5 qRT-PCR, the primer was obtained from Qiagen (Cat No. PPH01219F-200); GAPDH was used as control. For the profiler array, a total of 96 genes were profiled and data analysis was done using the Geneglobe portal on the Qiagen website. Samples (triplicates) were grouped into control (Non-Target) and test (CYP3A5 siRNA), and normalized with Beta-2-microglobulin (B2M) and Ribosomal protein, large, P0 (RPLP). A set of genes were identified based on fold change cutoff value of 2.0 and *p* value of 0.005. 

### 4.7. Luciferase Assay 

Cignal Lenti AR Reporter (luc) from Qiagen (Product No. 336851, Cat No. CLS-8019L) was used to generate an AR pathway sensing LNCaP and MDAPCa2b cell lines for the study of the AR signal transduction pathway. These lentivirus particles have androgen response elements (ARE) fused to luciferase, which detect any changes in AR downstream signaling. The cells were transfected according to the manufacturer instructions. Negative Control (only TATA box in place of ARE) transfected cell lines were also generated to measure background luciferase activity. The cells were maintained under puromycin selection pressure to select for stable chromosomal integration of the lentiviral constructs. The selected cells were tested for AR signaling pathway activation in response to DHT treatment after drug treatment using Bright-Glo Luciferase Assay from Promega (Cat No. E264A). The cells were collected into an Eppendorf tube and divided into two equal aliquots. One aliquot was used for luciferase assay and lysed with a Glolysis buffer (Cat No. E266A); the manufacturer’s guidelines were followed. The second aliquot was lysed with RIPA buffer for protein quantification.

### 4.8. Genotyping Assay 

DNA was isolated from cell lines using the QIAam DNA mini kit (Cat No. 51304) from Qiagen, according to the manufacturer’s instructions. TaqMan™ Drug Metabolism Genotyping Assay (Cat no. 4362691) from Applied Biosystems with 7500 Fast System was used to determine CYP3A5 *1 and *3 allelic status. The assay is a q-PCR based assay with two primer and two probe sets. One probe set identifies the wild type allele (*1) and is labelled with VIC dye (5’); the other identifies the *3 mutant and is labelled with the FAM dye (5’). Both the probes have a non-fluorescent quencher attached to the 3’ end, which increases the specificity of detection. The allelic differentiation probe/primer set has a minor groove binder (MGB); this modification increases the melting temperature (Tm) for a given probe length and results in greater differences in Tm values between matched and mismatched probes, which produces more robust allelic discrimination. After the end of the PCR reaction, the plate is read for the end point signal generated by both the reporter dyes (VIC and FAM), and allelic discrimination analysis is run to detect the specific alleles.

## 5. Conclusions

Based on our data, we suggest that taking CYP3A5 inhibitors concomitantly may clinically benefit patients undergoing ADT (enhancing its effect), whereas taking CYP3A5 inducers may reduce the efficacy of the ADT treatment (countering its effect). These observations suggest that the effect of these inhibitors and inducers may be more relevant in AA patients, as they tend to carry the wild type CYP3A5 and may result in therapeutic resistance. This study also suggests that care be taken while prescribing CYP3A5 inducers when patients are undergoing ADT. In addition, it also suggests that genetic testing for CYP3A5 polymorphism in patients may provide significant information about the potential impact of these interactions, facilitating personalized treatment regimens.

## Figures and Tables

**Figure 1 cancers-12-00989-f001:**
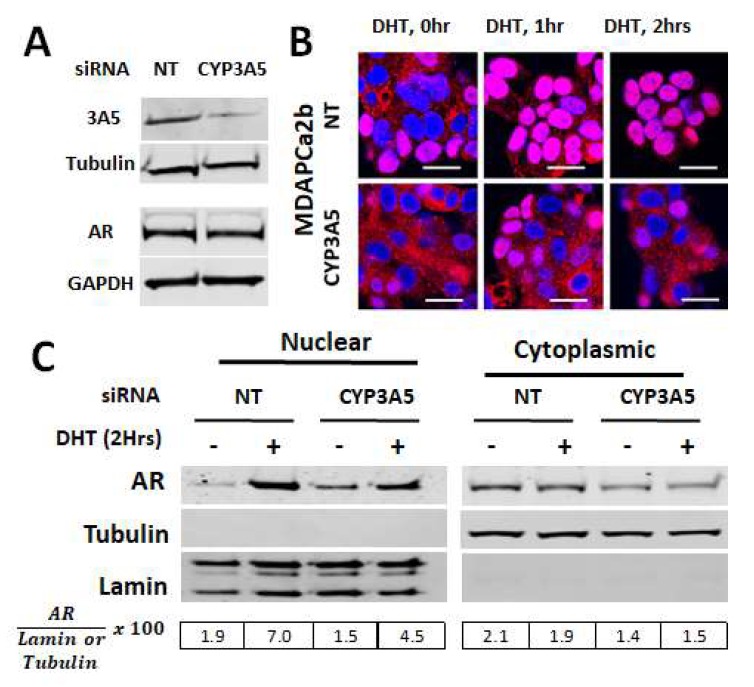
CYP3A5 siRNA downregulates AR (androgen receptor) nuclear translocation: (**A**–**C**) MDAPCa2b cells were transfected with CYP3A5 and non target (NT) siRNA. After 72 hours, the cells were given 10nM DHT treatment (0, 1, and 2 hours). (**A**) Western blot was performed to test CYP3A5 siRNA silencing efficiency at protein level using cytoskeletal fraction. Total protein was used to monitor changes in total AR protein expression. (**B**) For microscopy, the cells were labelled with AR primary antibody and Cy5 secondary (red) and nucleus was labeled with DAPI. The scale bar represents 50 µm. (**C**) After cell fractionation, western blotting was performed using cytoplasmic and nuclear fractions and probed for AR, Tubulin, and Lamin.

**Figure 2 cancers-12-00989-f002:**
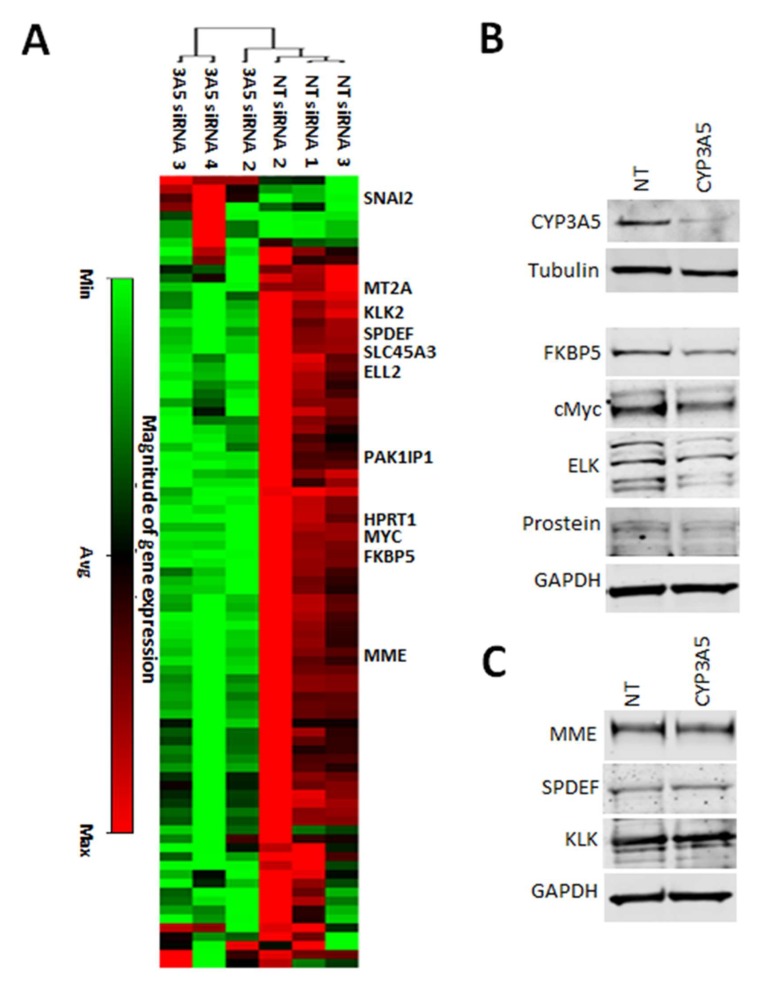
CYP3A5 siRNA downregulates expression of AR downstream regulated genes. (**A**) MDAPCa2b cells were seeded into 6-well plates. After 48 hours of allowing them to settle, the cells were treated with CYP3A5 siRNA or Non-Target siRNA and incubated for another 72 hours. RNA was isolated, followed by cDNA preparation, which was used in RT^2^ profiler assay. Fold-change values greater than 1 are indicated as positive- or an up-regulation (red) and less than −1 are indicated as negative or down-regulation (green). The *p* values are calculated based on a Student’s *t*-test of the replicate 2^ (−Delta CT) values for each gene in the control group and treatment groups. (**B**,**C**) Protein expression of the 7 genes that changes in gene expression was evaluated using western blotting. FKBP5, cMYC, ELK, and Prostein (SLC45A3) showed decreased protein expression in response to CYP3A5 knockdown, whereas, MME, SPDEF, and KLK showed no change.

**Figure 3 cancers-12-00989-f003:**
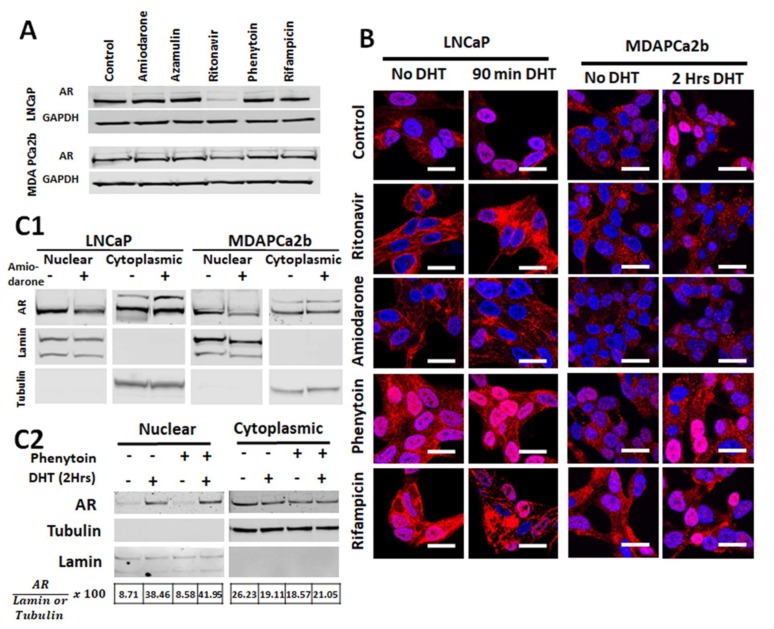
CYP3A5 inhibitors and inducers affect nuclear translocation of AR. (**A**) Effect of CYP inhibiting or inducing drugs on total AR expression. Total cell lysates from LNCaP and MDAPCa2b cells treated with CYP inhibitors (Ritonavir-35 µM, Azamulin 10 µM and Amiodarone-5 µM) and inducers (Phenytoin-50 µM and Rifampicin-30 µg/mL) for 48 hours was used for western analysis. (**B**) Nuclear localization of AR after CYP3A5 inhibitor/inducer treatment. Immunostaining was performed on LNCaP and MDAPCa2b cells that were treated with CYP3A inhibitors (Ritonavir-35 µM and Amiodarone-5 µM) and inducers (Phenytoin-50 µM and Rifampicin-30 µg/mL) for 48 hours in charcoal stripped serum media followed with and without DHT induction (90 min for LNCaP or 120 min for MDAPCa2b, 10 nM each). Nucleus is stained with DAPI (blue), AR is stained with Cy5-secondary (red) antibody. Scale bar represents 25 µm. A section from center of z-stack is shown here to demonstrate the localization of AR in the nucleus after treatments. (**C1**) Cell fractionation was performed after treating LNCaP and MDAPCa2b cells with Amiodarone (5 µM) for 72 hours. The cytoplasmic and nuclear fractions were evaluated using western. (**C2**) MDAPCa2b cells were treated with phenytoin (50 µM) followed by 10nM DHT induction (120 min); nuclear and cytoplasmic fractions were analyzed by western blotting. Lamin and tubulin are controls for nuclear and cytoplasmic fractions.

**Figure 4 cancers-12-00989-f004:**
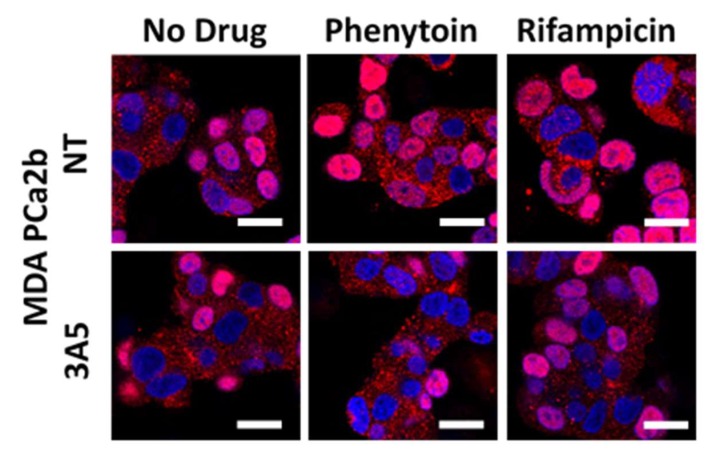
Modulation of AR signaling is dependent on CYP3A5 modulation. AR nuclear translocation by CYP3A inducers in NT/CYP3A5 siRNA treated MDAPCa2b cells. MDAPCa2b cells were treated with NT/CYP3A5 siRNA pool for 24 hours and then incubated with CYP3A inducers, phenytoin (75 µM), and rifampicin (30 µg/mL) for 48 hours in complete media. Confocal microscopy was performed and center of Z-stack is shown for nuclear AR localization. AR-red (Cy5) and nucleus (blue-DAPI). Scale bar represents 25 µm.

**Figure 5 cancers-12-00989-f005:**
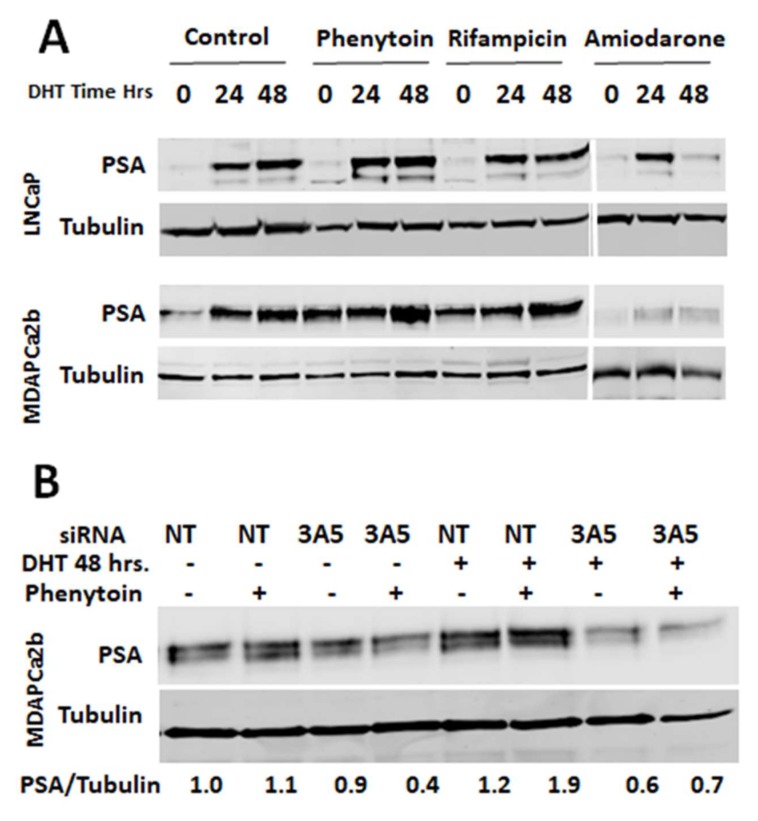
Modulation of PSA expression after CYP3A5 inhibitor/inducer treatment. (**A**) To confirm the effect of CYP3A5 modulating drugs on AR downstream signaling LNCaP and MDAPCa2b cells were treated with Phenytoin (50 µM), Rifampicin (30 µg/mL), and Amiodarone (5 µM) in charcoal stripped serum media followed by 24 or 48 hours of DHT (10nM) treatment. Total cell lysate was used to check PSA production using western analysis. (**B**) To confirm that the effect of CYP3A5 inducer on AR downstream signaling is due to its effect on CYP3A5, the PSA was measured in a CYP3A5 negative background (CYP3A5 siRNA). Cells were treated with non target (NT) or CYP3A5 siRNA pool. After 48 h of siRNA treatment, cells were treated with phenytoin (50 uM) and DHT (10 nM) for another 48 h, as indicated. The experiment was performed in charcoal stripped phenol red free serum media.

**Figure 6 cancers-12-00989-f006:**
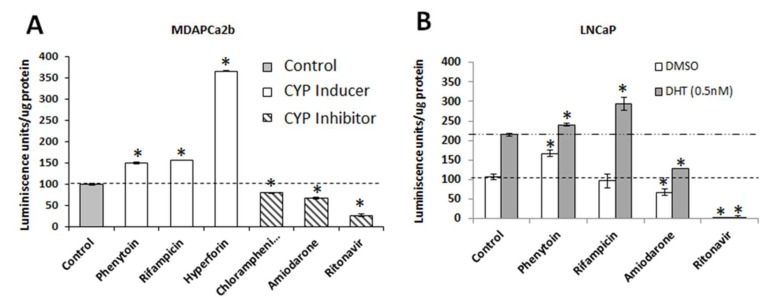
Reporter assay showing effect of CYP3A5 inhibitor/inducer treatment on AR downstream signaling. (**A**,**B**) MDAPCa2b and LNCaP cells transfected with androgen response elements (ARE) fused to luciferase were used to evaluate AR downstream signaling activity. MDAPCa2b cells were treated with known CYP3A5 inducers (Phenytoin-50 µM, hyperforin-200 μg/mL, and Rifampicin-30 µg/mL) and inhibitors (Ritonavir-35 µM, Azamulin-10 µM, chloramphenicol-10 µM, and Amiodarone-5 µM). LNCaP cells were treated with CYP inducing and inhibiting drugs in charcoal stripped serum followed by DHT (0.5 nM) induction for one hour. In both cases, CYP3A inducers showed increased AR signaling activity whereas CYP3A inhibitors showed decreased AR signaling activity. * *p* ≤ 0.05.

**Figure 7 cancers-12-00989-f007:**
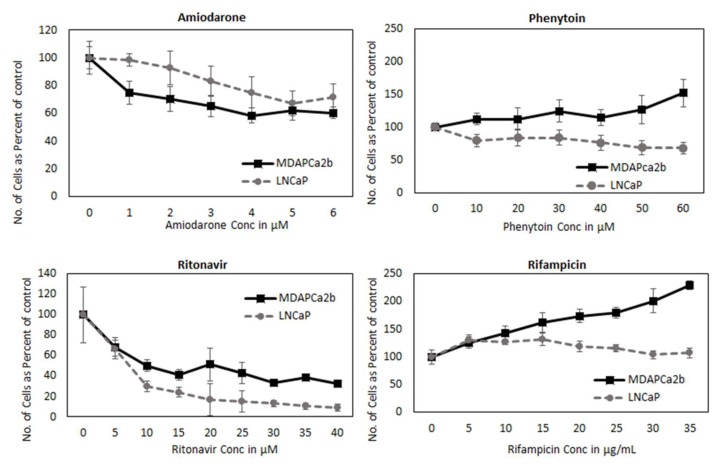
Effect of CYP3A5 inhibitor/inducer treatment on prostate cancer cell growth. LNCaP and MDAPCa2b cells were treated with a CYP3A inhibitors, amiodarone (0–6 µM) and ritonavir (0–40 µM); and CYP3A inducers Phenytoin (0–60 µM) and Rifampicin (0–35 µM) for 96 hours. The cell growth was accessed using MTS assay.

**Table 1 cancers-12-00989-t001:** CYP3A5 polymorphism analysis of commonly used prostate cancer cell lines.

Cell Line	Genotype	Origin
LNCaP	*3/*3	NHWA
22RV1	*3/*3	NHWA
C4-2	*3/*3	NHWA
E006AAhT	*3/*3	NHWA
MDAPCa2b	*1/*3	AA
RC77 T/E Tumor	*1/*3	AA
RC77 N/E Normal	*1/*3	AA

Seven androgen responsive prostate cell lines were tested for presence of wild type (*1) or mutant (*3) CYP3A5 polymorphism by using a qPCR-based genotype assay. NHWA-Non-Hispanic White Americans, AA- African Americans. RC77 T/E and N/E cell lines are derived from the same patient.

**Table 2 cancers-12-00989-t002:** CYP3A5 inhibition downregulates AR (androgen receptor) downstream regulated genes.

Gene Symbol	Fold Regulation	*p* Value
SLC45A3	−4.56	0.002
FKBP5	−4.43	0.002
MYC	−3.68	0.001
MME	−3.34	0.016
PAK1IP1	−3.25	0.016
ELL2	−3.25	0.004
KLK2	−2.82	0.009
HPRT1	−2.65	0.005
SPDEF	−2.58	0.012
MT2A	−2.45	0.001
SNAI2	3.32	0.005

Table showing downregulation of AR downstream genes with CYP3A5 siRNA treatment in MDAPCa2b cells. The fold change is in comparison with NT (non-target) pool siRNA treatment. The fold changes of all tested genes present in the array is listed in Appendix A.

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
