# Peer review of "Role of CYP3A5 in Modulating Androgen Receptor Signaling and Its Relevance to African American Men with Prostate Cancer"

_cancers, 2020, doi:10.3390/cancers12040989_

Round 1

Reviewer 1 Report

Identifying novel determinants of androgen receptor (AR) function is important for developing new therapies for prostate cancer, the disease which is driven by the activation of AR. The study by Gorjala identifies CYP35A5 factor that modulates androgen receptor activity, and also determines the relevance of CYP35A as an under-pinning factor causing differences between African-american and white populations (regarding the outcome of AR-targeted therapies). The authors provide data showing differential expression of CYP3A5 between African American (AA) and Non-Hispanic White-American origin AR-positive prostate cancer cell lines. Whereas 100 percent White-American cell exhibited polymorphism in CTP35A, AA-cell did not display it in all lines. The data shows that down-regulating CYP3A (by siRNA) or inducing CYP3A (by chemical agents such as Phenytoin) could modulate AR-regulated gene in metastatic cells lines suggesting an association between CYP3A and AR activity. The modulation of CYP3A also caused an effect on growth of cancer cells, particularly of AA-origin. The data is presented in an excellent manner. However the discussion is a bit longer and could be made succinct. In addition, role of  CYP3A inducers should be discussed with respect to prostate cancer therapy (because authors made a provocative conclusion about the harmful effects of CYP3A inducers; which are currently in clinical use). 

Author Response

Q1: However the discussion is a bit longer and could be made succinct. In addition, role of  CYP3A inducers should be discussed with respect to prostate cancer therapy (because authors made a provocative conclusion about the harmful effects of CYP3A inducers; which are currently in clinical use). 

A1: We thank the reviewer for the important suggestions; we have removed the redundancy from the discussion to make it crisp.  Although our data are highly suggestive and provocative, we recognize that they are preliminary proof of concept and may not be generalizable in the clinical setting. We plan to conduct prospective clinical studies to determine the clinical impact of concomitant prescription medications in men with prostate cancer undergoing ADT, we have added the information in the discussion (pg 14, line 430-434)

Reviewer 2 Report

The findings of this paper is very interesting and I recommend it for publication.

The only issue I have is the criteria for the choice of CYP3A5 inhibitors and inducers used in the experiments. The authors did not give any strong reference from any study (either case study reports or meta analysis that indicated that the selected drugs either enhanced the progression from androgen dependent to androgen independent (CRPC) or retard this transition.

Also authors did not show any evidence (reference) that any of the inducers or inhibitors are frequently prescribed to AA prostate cancer patients.

Author Response

Q1: The only issue I have is the criteria for the choice of CYP3A5 inhibitors and inducers used in the experiments. The authors did not give any strong reference from any study (either case study reports or meta analysis that indicated that the selected drugs either enhanced the progression from androgen dependent to androgen independent (CRPC) or retard this transition.

Also authors did not show any evidence (reference) that any of the inducers or inhibitors are frequently prescribed to AA prostate cancer patents.

A1: We thank the reviewer for their review and suggestions. This study is an initial proof of concept study to test the hypothesis that since CYP3A5 promotes AR activation it may alter treatment efficacy if the patients undergoing ADT concomitantly take CYP3A5 inducers/inhibitors. Since it is an initial study, only strong CYP3A5 inducer and inhibitors were used in the experiments. We were not able to find a meta-analysis or case report which provides the details of concomitant drugs prescribed along with treatment regimens.  Additionally, we did not find published  studies which provide the genotyping analysis of CYP3A5 alleles to test our hypothesis.

Among the drugs used for our study amiodarone, is a commonly prescribed drug for cardiac arrhythmias but we do not have data in reference to AA patients. Phenytoin is prescribed for epilepsy but its use is restrictive to only specific patients. To test the effect of CYP3A5 inducer and inhibitors in a clinical setting we are designing a pilot study with Fluoxetine (a CYP3A5 inhibitor) prescribed to patients with depression (common in men undergoing ADT) and test if it can improve ADT efficacy.

Reviewer 3 Report

As African American men carry the wild type CYP3A5 and express high levels of the CYP3A5 protein, this study investigates the role of CYP3A5 in modulating androgen receptor signaling using a prostate cancer cell line from an African American patient. The research question is interesting, but several weaknesses should be addressed.

  1. The data for the genotyping of CYP3A5 alleles in Table 1 is not shown. How reliable and accurate is the TaqMan assay for the genotyping of the prostate cancer cell lines? Could these results be confirmed by another method? A minor note is that the RC77 T/E lines are derived from the same patient, which should be mentioned.

  1. A critical missing piece of information seems to be that the authors do not show CYP3A5 mRNA or protein expression in MDAPCa2b cells, so it is not clear how this compares to CYP3A5 mRNA and protein expression in LNCaP cells.

  1. The authors do not directly show that CYP3A5 siRNA treatment reduces CYP3A5 mRNA or protein expression in MDAPCa2b cells.

  1. That the response to CYP3A5 inhibitors and inducers is more pronounced in MDAPCa2b cells compared to LNCaP cells is interesting, but it is not clear how generalizable this is to African American prostate cancer cell lines, since only 1 African American cell line was compared, and very few cell lines are available.

Minor comment:

In describing methodology, it would be helpful if the authors could clearly state the concentration of DHT used for each experiment, as this is not clear.

Author Response

We thank the reviewer for their critics and suggestions, we have tried to address all the queries to improve the manuscript.

Q1: The data for the genotyping of CYP3A5 alleles in Table 1 is not shown. How reliable and accurate is the TaqMan assay for the genotyping of the prostate cancer cell lines? Could these results be confirmed by another method? A minor note is that the RC77 T/E lines are derived from the same patient, which should be mentioned.

A1: The graphic representation of the allelic discrimination has been included as supplementary Fig 1 and one set of raw data has been included as a supplementary table 1 (pg 3, line98-99) . In the literature the frequently used assay for CYP3A5 allelic differentiation are q-PCR based assays with two separate primer sets. One primer set contains primer identical to the wild type and the other primer set contains primer identical to the mutant both, overlap the region of allelic differentiation. The second primer is same in both sets and is from a region with no mutation. The analysis is based on the difference of the Ct values from both reactions; the wild type allele has a lower ct value with the wild type specific primer and the mutant allele has a lower ct values with the mutated set, while the heterogeneous alleles have similar Ct values with both primer sets. In the taqman assay used for our experiments, it is multiplexed and is based on a primer/probe assay which is more specific than the commonly used q-PCR assay. The VIC dye is associated with the wild type allele where as the FAM dye is associated with the mutant (*3) allele.  The Taqman assay probes have the minor groove binding (MGB) modification which increases the melting temperature and results in greater differences in Tm values between matched and mismatched probes which provides more robust allelic discrimination. The details of the protocol has been added in the methods section to describe the assay (pg17, line 548-558).

We have included in the text that RC77 T/E and N/E are from the same patients (pg3, line 113).

Q2: A critical missing piece of information seems to be that the authors do not show CYP3A5 mRNA or protein expression in MDAPCa2b cells, so it is not clear how this compares to CYP3A5 mRNA and protein expression in LNCaP cells.

A2:  The experiments showing comparison of mRNA and protein expression between the LNCaP and MDAPCa2b cells has now been included in the manuscript (Supplementary figure 2). Both MDAPCa2b and LNCaP lines express similar levels of CYP3A5 mRNA. The MDAPCa2b cells express more functional protein as compared to LNCaP cells as shown in Supplementary figure 2B (Pg 3, line 105-107). The antibody used only recognizes full-length protein, the truncated inactive protein generated in *3 allele is not recognized by this antibody.

Q3: The authors do not directly show that CYP3A5 siRNA treatment reduces CYP3A5 mRNA or protein expression in MDAPCa2b cells.

A3: We have incorporated data showing the reduction of CYP3A5 mRNA and protein in our edited manuscript. The CYP3A5 siRNA pool reduces both CYP3A5 mRNA (Supplementary Figure 3) and protein levels (Figure1 and 2) (pg 4, 122-124 and pg 5, 150-152) . Unpublished data in our laboratory show that CYP3A5 in the prostate cancer cells co-localize with tubulin involved in AR nuclear localization. Hence, our RIPA buffer was not able to extract the CYP3A5 from the cells. We performed cell fractionation experiments and have provided western blot using cytoskeletal fraction indicating reduction in CYP3A5 protein expression.

Q4: That the response to CYP3A5 inhibitors and inducers is more pronounced in MDAPCa2b cells compared to LNCaP cells is interesting, but it is not clear how generalizable this is to African American prostate cancer cell lines, since only 1 African American cell line was compared, and very few cell lines are available.

 A4: I agree with the reviewer that due to limited availability of African American prostate cancer cell lines we were not able to perform the comparison with multiple lines. We only have access to gDNA from RC77 T/E and N/E lines from Dr. Rhim so we cannot use it for other studies.

Minor comment:

Q5: In describing methodology, it would be helpful if the authors could clearly state the concentration of DHT used for each experiment, as this is not clear.

A5: We have included the DHT concentrations with each experiment in the legend section.

Once again thank you for your time and constructive suggestions.

Best Regards,

Ranjana

Round 2

Reviewer 3 Report

The new information provided has significantly strengthened the paper.